# Electromagnetic and Calorimetric Validation of a Direct Oil Cooled Tooth Coil Winding PM Machine for Traction Application

**Alessandro Acquaviva ***, **Stefan Skoog** , **Emma Grunditz** and **Torbjörn Thiringer**

Department of Electrical Engineering, Chalmers University of Technology, Hörsalsvägen 9-11, 41296 Gothenburg, Sweden; stefan.skoog@chalmers.se (S.S.); emma.grunditz@chalmers.se (E.G.); torbjorn.thiringer@chalmers.se (T.T.)

*** Correspondence: alessandro.acquaviva@chalmers.se; Tel.: +46-73-5704-363

**Abstract:** Tooth coil winding machines offer a low cost manufacturing process, high efficiency and high power density, making these attractive for traction applications. Using direct oil cooling in combination with tooth coil windings is an effective way of reaching higher power densities compared to an external cooling jacket. In this paper, the validation of the electromagnetic design for an automotive 600 V, 50 kW tooth coil winding traction machine is presented. The design process is a combination of an analytical sizing process and FEA optimization. It is shown that removing iron in the stator yoke for cooling channels does not affect electromagnetic performance significantly. In a previous publication, the machine is shown to be thermally capable of 25 A/mm$^2$ (105 Nm) continuously, and 35 A/mm$^2$ (140 Nm) during a 10 s peak with 6 l/min oil cooling. In this paper, inductance, torque and back EMF are measured and compared with FEA results showing very good agreement with the numerical design. Furthermore, the efficiency of the machine is validated by direct loss measurements, using a custom built calorimetric set-up in six operating points with an agreement within 0.9 units of percent between FEA and measured results.

**Keywords:** permanent magnet machines; electromagnetic design; model verification; performance evaluation; calorimetric measurement; oil cooling; efficiency validation

## 1. Introduction

In recent years, research and development of automotive electric traction machines has greatly intensified. Apart from high efficiency and low cost, a specific design target for these machines is high power density (i.e., power per volume) [1–3]. Aiming for high power density means maximizing the material utilization of the machine, which is essential to achieve cost-effective solutions for mass-production, and low package volumes that enables effective system packaging.

Therefore, high-power density electric machines as well as power electronics are seen by the US department of energy as critical enablers of large-scale electric vehicle adoption [2]. The previous technical target specified in the US DRIVE Technology Roadmap for 2020 was 5.7 kW/liter for a 55 kW peak machine. However, the latest target for 2025 is 50 kW/liter for a 100 kW peak machine. Even though existing solutions are well positioned for the 2020 target, significant challenges lies ahead to live up to the next generation of traction systems in year 2025. This is exemplified in Table 1, which provides a comparison of power densities, both net (total volume of active steel) and gross (volume of complete electric machine casing), for state-of-the-art automotive electric traction machines.

High-power density electric machines can be achieved by utilizing factors such as:

- high mechanical speed [3], by a high electric frequency, or high pole number [4]
- high airgap flux density (i.e., magnetic loading), e.g., by using high energy magnets, or a field winding excitation, both in combination with core material with saturation at high flux density [5]
- high current loading, or high current density, while simultaneously assuring a low thermal resistance between the winding and coolant

For the sake of high magnetic loading, the development of electric drive-trains is dominated by the permanent magnet synchronous machine (PMSM) [6], characterized by its high power density as well as high efficiency [1,7,8]. However, as identified in [2], to reach even higher power density, more research is required regarding "improved thermal materials", as well as "advanced cooling/thermal management techniques to reduce size, cost and improve reliability". Extensive engineering efforts are devoted to solve these challenges, as summarized well in [9–11]. An apparent aim is to try to bring the coolant medium closer to the main sources of heat, i.e., the stator core and winding, as opposed to so called cooling jackets.

One possibility is then to use a tooth-coil winding machine (TCWMs), also known as non-overlapping fractional slot concentrated winding (FSCW) machine. With this machine type, it may be possible to devote some of the space that is normally used for active material, for cooling channels instead, without sacrificing performance. Still, it offers high torque density and high efficiency [12–14] when combined with a permanent magnet (PM) rotor, as well as low cost manufacturing. In [15], a 12-slot 8-pole TCWM for traction application is compared with a distributed winding (DW) interior-magnet, a switched reluctance, and an induction machine. The TCWM is shown to perform best in terms of torque density, even without considering the shorter winding overhang. Efficiency-wise, the two PM machines are comparable, TCWM being slightly better in the low speed region and less efficient at high speeds compared to the DW machine. Other interesting traction motor designs using TCWM are presented in [16–20], however, without the integration of direct cooling in the stator, continuous current densities above 20 A/mm$^2$ are hardly reached, which limits the torque density.

Previous proposals of high performing liquid cooling techniques for TCWMs comprise examples such as the following:

- using conductive pipes in the slots, with the drawback of generating large eddy current losses [21]
- theoretical evaluation of the concept of flushing the entire stator and rotor with oil coolant [22]
- direct-water cooled coils by winding a coolant carrying steel pipe with litz wire, validated in a 205 kW machine for a bus application [23]
- using fluid guiding structure and airgap sealing to allow for oil cooling within the slots [24].

Neither of these examples though, have reached the levels of power density and efficiency that is presented in this paper, nor have they experimentally verified as high current densities.

Enabling high current density, which in turn means high copper losses at peak operation, does not preclude the traction machine to achieve high energy efficiency. The reason is that during driving, the machine operates most of the time in part-load, i.e., the low torque region, as shown for several drive cycles in [25]. Achieving high energy efficiency can significantly extend the range of the vehicle for a given battery pack.

This purpose of this paper is to present the electromagnetic design and verification of a high-power density, high-efficiency 50 kW TCWM, sized for traction application of passenger vehicles. The high power density is achieved by integration of direct-oil cooling in the stator yoke as well as in the slot via a potting material with high thermal conductivity, which shapes the cooling channels. The details of the design and verification of the cooling solution are presented by the authors in [26]. This paper instead focuses on the electromagnetic design validation. Particularly, it is shown how the placement of stator yoke oil cooling channels can be done without affecting the electromagnetic performance, which has not been found in literature. Experimental validation of the torque, no-load EMF and inductance present a very good match with simulations. Furthermore, the efficiency is verified through direct

loss measurement in a custom-made calorimetric set-up, also showing a good match with simulated results. Finally, the paper guides the reader through the main choices and steps for the design of a traction PM-TCWM intended for high volume production.

**Table 1.** Performance metrics for traction machines.

| Machine | Net Power Density (kW/L) | Gross Power Density (kW/L) | Peak Efficiency |
|---|---|---|---|
| Toyota HSD 2010 [27] | 22 | 5.2 | 96% [27] |
| BMW i3 [3] | 20 | 9.2 | 94% [28] |
| Machine presented here | 19 | 6.7 | 95% |
| DoE Target 2020 [2] | - | 5.7 | - |
| DoE Target 2025 [2] | - | 50 | - |

## 2. Machine Design

The target application of the machine in this paper is a traction motor for either a small passenger vehicle, or an assist motor in a plug-in hybrid electric vehicle. A fixed-gear reduction gearbox is assumed between the traction machine and the vehicle wheel pairs, to allow high-speed operation. The design specifications are listed in Table 2.

**Table 2.** Input design specifications for the electric machine.

| Quantity | Symbol | Value | Unit |
|---|---|---|---|
| Peak torque | $T_{max}$ | 140 | Nm |
| Peak power | $P_{max}$ | 50 | kW |
| Base speed | $n_b$ | 3600 | rpm |
| Max speed | $n_{max}$ | 11,000 | rpm |
| Coolant max temperature | $\Theta_{max,c}$ | 60 | °C |
| Max winding temperature | $\Theta_{max}$ | 180 | °C |
| DC bus voltage | $V_{DC}$ | 600 | V |

### 2.1. Electromagnetic Analytical Sizing and Design Choices

The electromagnetic design is based on an analytical sizing method combined with a finite element mapping and verification. Given the target power rating, a desired high power density, and the ability to gear the machine, an axially long and radially small machine is selected to achieve high mechanical speeds. A fundamental frequency of 1.2 kHz is assumed at maximum speed, limited by machine iron losses and the maximum switching frequency (typically 10 kHz) together with a reasonable frequency modulation index of a high-performance traction inverter. A pole-pair number of five or six will now set the maximum mechanical speed to 14.4 and 12 krpm, respectively. Using the methodology presented in [13], the only slot (Q)-pole (p) options left for a balanced three-phase machine are Q6p8, Q12p8, Q12p10 and Q9p12, all illustrated in Figure 1. A FEA comparison between these four feasible pole-slot combinations in terms of output torque and iron losses at 300 Hz electrical frequency is shown in Figure 2. This comparison is done keeping the main geometrical parameters (material types, axial length, stator external and internal diameter, airgap length, tooth embrace) and loading parameters (current density, airgap flux density) constant in all the designs at the values listed in Table 3. The Q12p10 machine presents the highest torque density and lowest iron losses among the machines compared, as can be seen in Figure 2. The Q12p10 offers the best balance between high fundamental winding factor and low harmonic leakage inductance factor; 0.933 and 0.97 respectively. More high-level performance indices are derived in [13].

The analytical sizing procedure, presented in [29], is based on an optimization of the split ratio; the ratio between internal to external stator diameter. The analytical model is built by writing the torque equation of the brushless PM machine as a function of the main geometrical parameters and

the two main loading parameters: Airgap flux density and current density. It is shown in [29] that the choice of the split ratio is a trade-off between efficiency and torque density. In the design presented in this paper the split ratio for maximum efficiency is 0.62. The choice is also driven by thermal considerations; a low value of split ratio leads to a high copper and slot area, which, for a fixed current density, means critical cooling requirements. A stator design without tooth tips is chosen to improve manufacturability by allowing coils to be pre-wound separately before radial insertion onto the stator teeth. The rotor is chosen as an internal V-shaped PM with air barriers, which offers high saliency and enables a large field weakening (FW) area. Embedded rotor magnets, as opposed to surface mounted ones, also reduce magnet losses caused by harmonics in the airgap MMF [30,31]. One additional measure taken to limit excessive magnet losses is magnet segmentation [31,32] by using 20 equal Vacodym 745DHR NdFeB magnet units stacked axially in each rotor slot. The lamination used is M235-35 with a thickness of 0.35 mm which has been laser-cut for the prototyping.

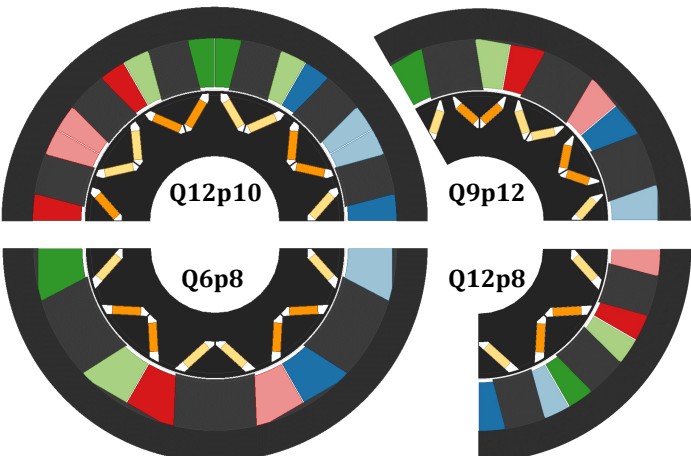

**Figure 1.** Smallest unique section of rotor and stator geometries for the four tooth-coil winding machine (TCWM) slot-pole combinations evaluated through FEA.

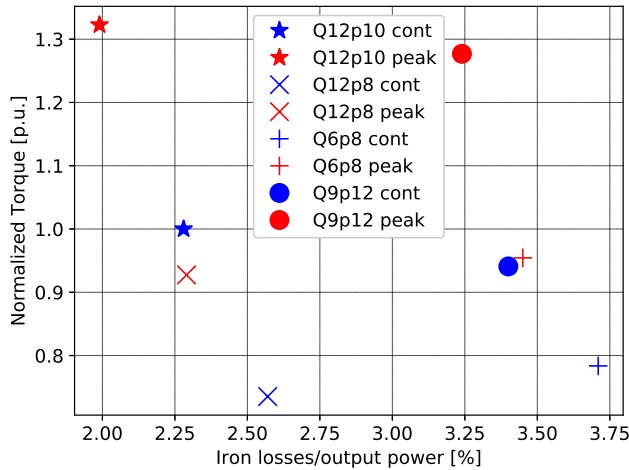

**Figure 2.** Numerical comparison of the different pole slot combinations for peak and continuous operation at 300 Hz electrical frequency. All machines have the same volume, airgap flux density and current density ($A_{cont}$ and $A_{peak}$), see Table 3. The torque is normalized to the Q12p10 machine in continuous operation.

The resulting machine parameters are listed in Table 3. The maximum phase current corresponds to 35 A/mm$^2$ current density in the copper conductors. The coil disposition and geometries of

stator and rotor with details about disposition of conductors and cooling channels are shown in Figure 3. The Q12p10 machine has a key winding factor [13] of 2, meaning each phase coil consists of two electrically series connected coils on adjacent teeth. Each coil has 28 turns, which allows for a 1.6 mm diameter enamel copper wire to be used and avoiding parallel strands in the coils. Having a bobbin which can be inserted, limiting the conductor diameter and avoiding parallel strands, enables the use of a linear winding machine, which can drastically reduce the manufacturing cost at high volume production. Each set of two series coils are then parallel connected to form a full phase winding, as shown in Figure 4.

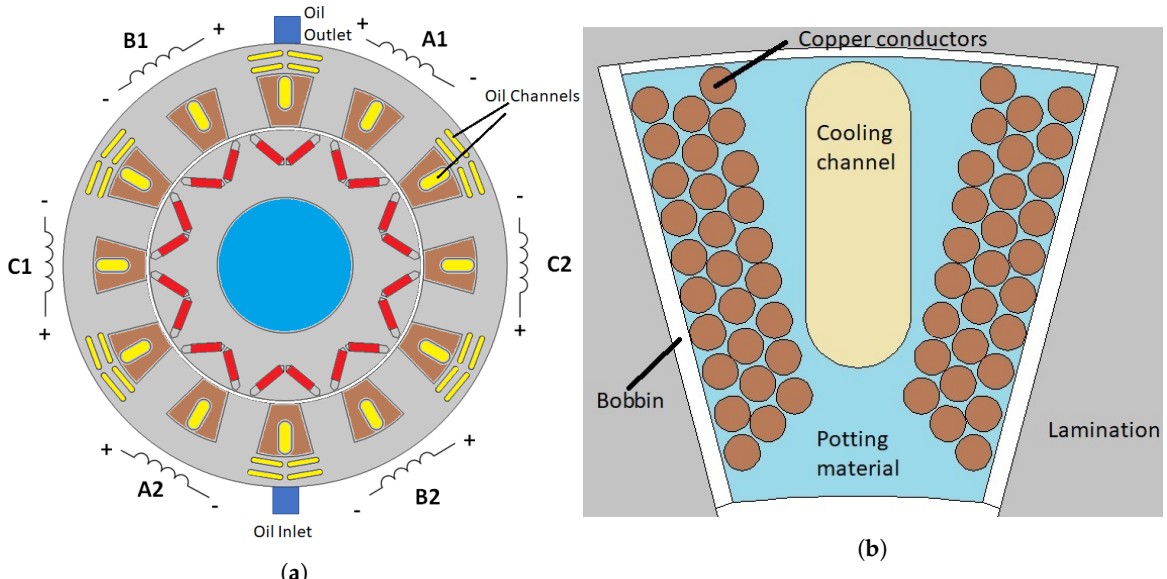

**Figure 3.** Details of lamination geometry, cooling channels and conductor disposition. (**a**) Stator and rotor laminations geometry, coil disposition and cooling channels; (**b**) The arrangement of copper conductors, potting material and oil cooling channel within one slot.

**Table 3.** Resulting geometry and loading data from the machine dimensioning procedure.

| Quantity | Symbol | Value | Unit |
|---|---|---|---|
| Outer Stator diameter | $D_e$ | 180.0 | mm |
| Inner Stator diameter | $D$ | 111.4 | mm |
| Active length | $L$ | 100.0 | mm |
| Tooth width | $w_t$ | 17.0 | mm |
| Stator Yoke height | $h_{yoke}$ | 13.0 | mm |
| Magnet thickness | $h_m$ | 3.5 | mm |
| Diameter of each conductor | $d_N$ | 1.6 | mm |
| Number of turns per coil | $N$ | 28 | - |
| Airgap radius | $r_\delta$ | 0.70 | mm |
| Peak airgap flux density | $\hat{B}_\delta$ | 0.9 | T |
| Cont. RMS current | $A_{cont}$ | 25 | A/mm$^2$ |
| | $I_{cont}$ | 100 | A |
| Peak RMS current | $A_{max}$ | 35 | A/mm$^2$ |
| | $I_{max}$ | 140 | A |
| Net machine volume | $V_{net}$ | 2.57 | l |
| Gross machine volume | $V_{gross}$ | 7.43 | l |

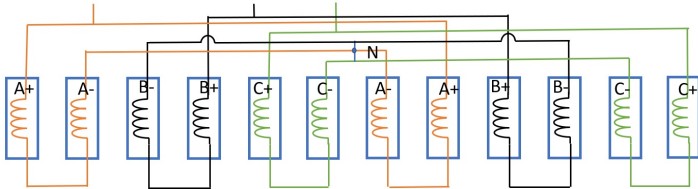

**Figure 4.** Winding disposition and connection. For this prototype machine, both parallel connection and Y connection is made outside the housing through six connection terminals.

The machine active parts and total active weight are recorded and presented in Table 4. The net power mass density for the machine is 3.39 kW/kg.

**Table 4.** Weight of active materials for the prototype machine.

| Stator Lam. | Rotor Lam. | PMs | Copper | Total | |
|---|---|---|---|---|---|
| 8.1 | 4.2 | 0.66 | 1.8 | 14.76 | [kg] |

## 2.2. Cooling Channel Size and Electromagnetic Effects

Introducing cooling channels in the stator back yoke by removing iron after the split ratio optimisation might affect the electromagnetic performance negatively by increasing the reluctance path for the rotor PM flux and/or the linked flux between stator coils. The Q12p10 machine features low mutual inductance between phases by linking the vast majority of the flux generated by phase windings in a loop contained within the two adjacent teeth belonging to the same phase group. This scenario is illustrated in the top part of Figure 5, without the remanence PM flux and 100 A in phase A. The change in self-linked flux due to magnetic saturation caused by the cooling channels positioned between the phases is negligible when channels are sized at $h_{bar}$ = 2.0 mm. Figure 5 shows that the flux density in the yoke with barriers is still well below saturation values. In [30], a similar yoke barrier strategy is used to partly block the flux from sub-harmonics which exist in this winding layout, resulting in a reduction in both iron and PM losses in the Q12p10 machine. The solution in [30] evaluates yoke barriers situated between the teeth within one phase, which leads to a large reduction of torque for the machine type preferred in our paper.

To find out the performance implications of flux barrier between the phases, including the PM flux, an FEA parametric sweep is performed. The cooling channels height $h_{bar}$ is swept from from 0.1 to 6.0 mm, corresponding to cooling channels occupying in total 2–92% of the stator back yoke height. The final design is using 2.0 mm barrier height, as shown in Figure 5. Average torque, torque ripple and iron losses are evaluated at 5000 rpm, from zero up to to rated peak current; 140 A (35 A/mm²). The results for the highest current, which has the most dramatic impact, are shown in Table 5. The average torque output is monotonically decreasing with increased barrier height. However, more than half of the yoke height (4.0 mm) can be cut out before 1% average torque loss is experienced. The relative torque ripple (pk2pk/avg) is kept robustly at 8% for all barrier sizes except the largest value. Moreover, the PM flux percentage reduction due to the introduction of barriers is negligible, also presented in Figure 5. Regarding iron losses, when seen relative to the mechanical output power ($P_{iron}/P_{mech}$), a small monotonically increasing trend can be seen of less than 0.1 unit of percent from the smallest barrier to the largest. Using dual 2 mm coolant barriers positioned between the phase groups is considered to have negligible impact on electromagnetic performance, and still offer enough cross-section area for low-viscosity oil to flow without significant pressure drop. The shape of the channels and the choice of having four parallel channels instead of one is driven by flow split evaluation as presented in [26]. The resulting pressure drop is measured to 37.6 kPa at the maximum oil flow of 6.0 l/min at room temperature.

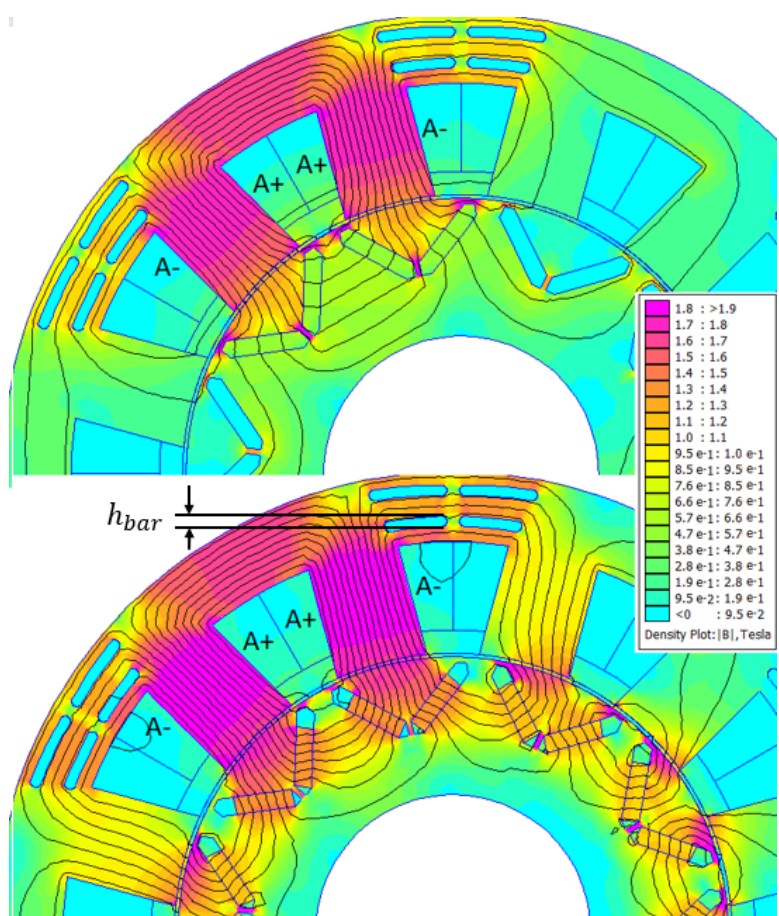

**Figure 5.** FEA established magnetic flux generated by 100 A in phase A with (bottom) and without (top) remanence flux in the magnets.

**Table 5.** Results from FEA evaluation of cooling barrier size at 5000 rpm, 140 A (35 A/mm$^2$), 120° current angle. $h_{bar}$ = 0.1 is selected as reference for relative changes. $h_{bar}$ = 2.0 is chosen for the prototype machine.

| Barrier Height | | Average Torque | | Torque Ripple | | Iron Losses | | PM Flux |
|---|---|---|---|---|---|---|---|---|
| $h_{bar}$ | $2h_{bar}/h_{yoke}$ | | | pk2pk | pk2pk/Avg | | $P_{iron}/P_{mech}$ | Change |
| (mm) | (%) | (Nm) | (±%) | (Nm) | (%) | (W) | (%) | (±%) |
| 0.1 | 1.5 | 145.66 | 0.00 | 12.34 | 8.47 | 980.2 | 1.29 | 0.00 |
| 1.0 | 15 | 145.62 | −0.03 | 12.51 | 8.59 | 986.5 | 1.29 | 0.00 |
| **2.0** | **31** | 145.59 | −0.05 | 12.24 | 8.41 | 999.0 | 1.31 | 0.00 |
| 3.0 | 46 | 145.35 | −0.21 | 12.25 | 8.43 | 1009.5 | 1.33 | −0.02 |
| 4.0 | 62 | 144.00 | −1.14 | 12.03 | 8.35 | 1021.3 | 1.35 | −0.17 |
| 5.0 | 77 | 140.04 | −3.86 | 11.43 | 8.16 | 1006.8 | 1.37 | −0.58 |
| 6.0 | 92 | 132.34 | −9.14 | 14.24 | 10.76 | 958.9 | 1.38 | −1.13 |

This type of utilization of part of the stator yoke to introduce cooling channels can be generalized for all TCWMs featuring a key winding factor equal to an even integer; which corresponds to an even number of adjacent tooth-coils belonging to the same phase winding [13].

### 2.3. Slot Fill Factor and Cooling Channels

The cooling channels are derived from the space in between the two coils, which is unused space in the slot. These are shaped in the epoxy material with Teflon sticks that are removed after the potting process. The potting material is an epoxy resin with an enhanced thermal conductivity at 1.9 W/(m·K). Each slot cross section area measures 350 mm$^2$, out of which in-slot cooling channels occupies 100 mm$^2$,

yielding 250 mm$^2$ for windings including insulation. The efective copper area is measured to 113 mm$^2$, resulting in a net fill factor of 0.45 and a bulk fill factor considering the total slot area of 0.32. The stator with the windings mounted before and after potting is presented in Figure 6. All thermal and cooling design considerations, including fluid dynamic analysis, for this machine are presented in [26].

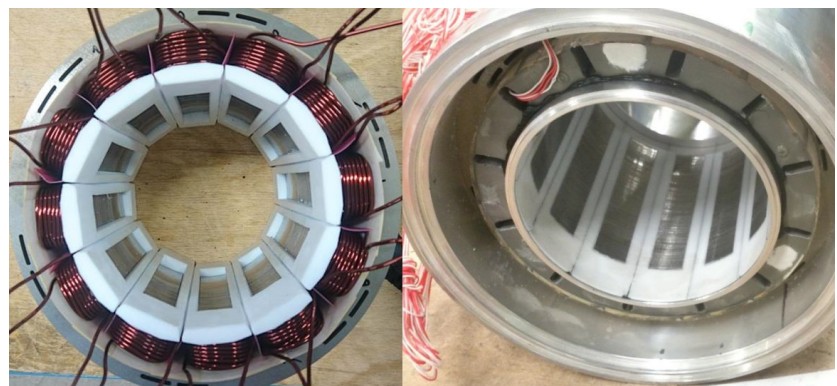

**Figure 6.** Wound stator before (left) and after (right) potting. The white plastics in the left picture are the bobbins used to pre-form the coils outside the machine. The excess material in the airgap is removed before potting and therefore not seen in right picture.

## 3. Numerical Evaluation

FEA is used to predict the electromagnetic performance of the machine, including loss- and flux maps. The losses are used as input for evaluating the thermal performance and the efficiency. Interior PM machines, when functioning as motors, are typically operated in the second quadrant of the d-q current plane. This is done in order to use both the PM flux generated torque, the reluctance torque and, at high speed operation, field weakening. Maximum currents for negative d-axis and positive q-axis are evaluated analytically and used as references to map the operating region. A set of 2D quasi-static magnetic FEAs over one electrical period are performed for each operating point in the mapping.

### 3.1. Loss Mapping

PM and iron losses for each operating point are evaluated by using the x and y components of the B-field ($B_x$ and $B_y$) for each mesh element of the rotor and stator iron. The magnetic vector potential $A$, which in 2D FEA has only the z component, is acquired for the PMs mesh elements. The iron losses are computed by performing the Discrete Fourier Transform (DFT) of $B_x$ and $B_y$ of stator and rotor iron elements. For each harmonic $i$ and mesh element $j$

$$B^2_{fe,i,j} = B^2_{x,i,j} + B^2_{y,i,j} \, . \tag{1}$$

The sum of all the contributions of the harmonics and all the $m$ elements [25] defines the total iron losses:

$$P_{fe,\omega} = K_m \sum_{j=1}^{m} \sum_{i=1}^{n/2} \frac{V_j}{k_s} \gamma_{fe} (k_e B^2_{fe,i,j} f_i^2 + k_h B^2_{fe,i,j} f_i) \, , \tag{2}$$

where $\gamma_{fe}$ is the iron mass density, $k_e$ and $k_h$ represent eddy current and hysteresis loss factors extracted from lamination data, $f_i$ is the electrical frequency at the $i^{th}$ harmonic, where the fundamental harmonic is the first harmonic present in the machine MMF for a certain angular speed $\omega$. $k_s$ is the stacking factor and $V_j$ is the volume of the mesh element $j$. Furthermore, $K_m$ is a iron loss scaling coefficient. This coefficient accounts, in the first place, for the effect of laser cutting of non-oriented electrical steel causing structural changes at the cutting edge, which finally affect the magnetic properties. Secondly,

there is often a mismatch between the loss data found in the steel data sheets and the measured values when implemented in the machine.

Copper and iron losses are calibrated with the measured phase resistance, and by measuring the iron losses with the calorimetric set up described in Section 4, at 3000 rpm and no load, resulting in an iron loss scaling coefficient $K_m$ = 2.0 (see Equation (2)). This means measured iron losses are twice as large as the FEA estimated losses when using interpolated loss coefficients from the soft iron manufacturer. Typical values of iron loss scaling coefficients found in literature are in the range between 1.5 and 2.0 [22]. The central part of the slot, closest to the airgap, is generally the most critical when it comes to AC losses. In the proposed machine this area is not filled with conductors, Figure 3b. Also, the skin dept at the maximum fundamental frequency 1.2 kHz is 1.9 mm which is significantly higher than the copper conductor radius 0.8 mm. For these reasons the AC losses are considered negligible.

PM losses are calculated by applying DFT to the vector potential *A* for all mesh elements belonging to the PMs. The induced current density in PMs is calculated as

$$J_{PM} = -\frac{1}{\rho_{pm}}\frac{dA_{PM}}{d\theta}\omega + C(t)\,, \tag{3}$$

where $\rho_{pm}$ is the resistivity of the magnet material and $C(t)$ is an integration constant ensuring the net total current flowing inside each magnet is zero at any time instant. The method is fully described in [33]. PM segmentation in the axial direction is accounted for by considering an equivalent eddy current resistance path. A flow chart summarizing the loss mapping procedure is shown in Figure 7. The torque-speed maps showing the different loss contributions are presented in Appendix A.

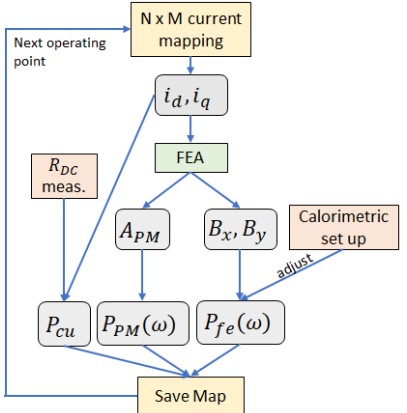

**Figure 7.** Loss mapping procedure flow chart.

*3.2. Inductance and Linked PM Flux*

The d and q-axis inductance as a function of current are useful parameters for control purpose. The flux linkage of each phase and for all operating points is known from the FEA. Using the Clark and Park transformation the d-axis and q-axis flux linkage are derived. The PM flux linkage can be calculated at $i_d = 0$ as a function of the q-axis current as

$$\Psi_{pm}(i_q) = \Psi_d\Big|_{i_d=0}. \tag{4}$$

The d-axis and q-axis inductance can be computed as

$$L_d = \frac{\Psi_d - \Psi_{pm}}{i_d} \tag{5}$$

$$L_q = \frac{\Psi_q}{i_q}. \tag{6}$$

The inductance maps as a function of the d-axis and q-axis currents are presented in Figure 8.

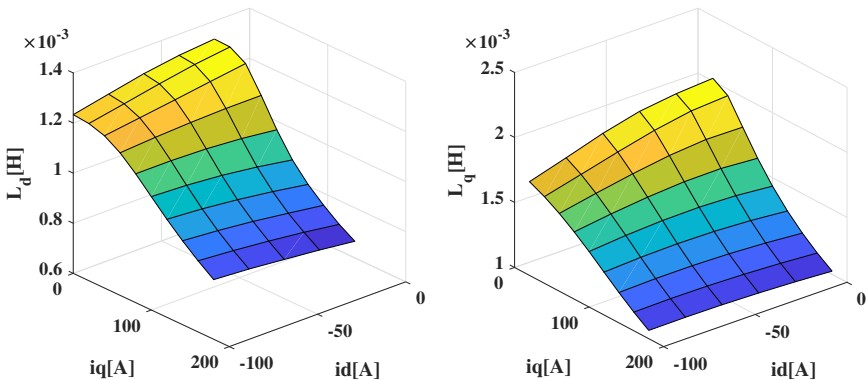

**Figure 8.** Inductance maps established through FEA. On the left the d-axis inductance and on the right the q-axis inductance.

### 3.3. Induced Voltage

The winding induced phase voltage is calculated by applying the derivative of the flux with respect to the angle, using approximation of derivatives by finite differences, as

$$V_{a,b,c} = \frac{\Delta\Psi_{a,b,c}}{\Delta\theta}\omega + R_{DC}\,i_{a,b,c}\ , \tag{7}$$

where $\Delta\Psi_{a,b,c}$ is the finite difference in flux linkage for a mechanical angle step of $\Delta\theta$, is the $R_{DC}$ is the measured DC phase resistance, $\omega$ is the mechanical rotational speed and $i_{a,b,c}$ is the phase current.

### 3.4. Torque

The torque is computed through Maxwell stress tensor calculation. This is calculated through integration paths, in circular bands, in the air gap and averaged to improve the accuracy by cancelling out some numerical noise due to the differentiation. This is referred to as Arkkio's method [34]. Thanks to the salient rotor structure, the machine presents a significant amount of reluctance torque at the optimal MTPA angle. The PM and reluctance torque contributions for different current densities in the low speed region is presented in Figure 9.

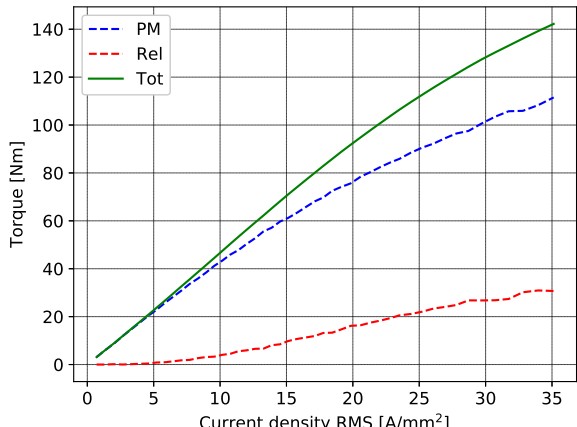

**Figure 9.** Permanent magnet (PM), reluctance and total torque when operating with optimal MTPA angle as a function of copper RMS current density.

## 4. Test Set up

In order to evaluate the performance of the assembled traction machine, a calorimetric set up is designed and constructed. The lab environment consists of the prototype machine, power electronics, an electric dynamometer, a custom made oil-to-water cooling system and a dSpace SCALEXIO-rack. The setup is illustrated in Figure 10 and shown on the photo in Figure 11.

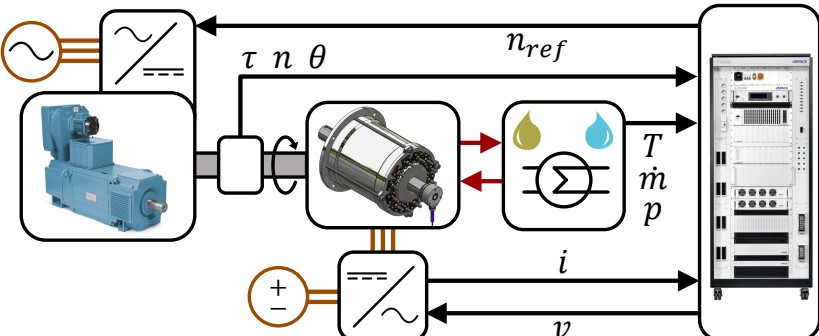

**Figure 10.** Calorimetric measurement system setup: A dSpace system is controlling the dynamo, the prototype machine, and the custom made oil-to-water cooling system.

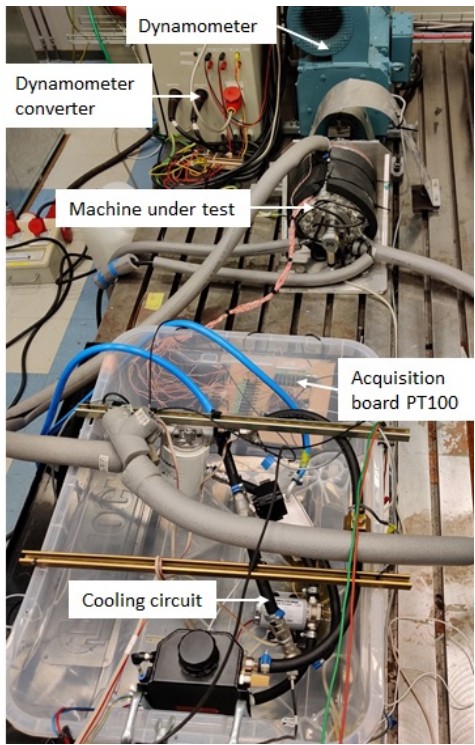

**Figure 11.** Picture of calorimetric test setup, showing the thermal insulation of the machine, oil hoses, and phase cables.

Sensors are installed and calibrated to measure DC and AC voltage ($v$), phase currents ($i$), output torque ($\tau$), rotational speed ($\omega$), rotational position ($\theta$), oil and water flow-rates ($\dot{m}$), oil and water inlet and outlet temperatures ($T$) and pressures ($p$), water flows and pressures. Critical measurement equipment is listed in Table 6. The dSpace rack acts as the control hub of the dynamo setup. With the total setup, both electrical, mechanical and direct-loss power can be measured and displayed continuously and reference values can be set via a PC GUI. Oil and water temperatures are measured and converted to digital form with high resolution. The inlet-to-outlet temperature drop ($\Delta T$) of the coolant oil is established. Together with the oil mass flow ($\dot{m}$), oil density ($\rho = 821$ kg/m$^3$) and the oil specific heat capacity ($c_p = 2100$ J/kg/K) the total loss heat transported out of the machine can be calculated as

$$P_{cal} = \dot{m}\,\rho\,c_p\,\Delta T\,. \tag{8}$$

More information on the oil used can be found in [26]. The test machine and the oil system is thermally insulated towards the ambient in order to minimise thermal leakage and enable accurate calorimetric measurements. A glass fiber washer acts as a thermal insulator between the machine enclosure and the mounting frame. The coolant oil is heat exchanged with mass flow controlled tap water so that the inlet oil temperature to the machine is kept at 22 °C, in order to minimize leakage heat from the surroundings.

A three-leg IGBT based inverter, operating at 5 kHz SVM closed-loop current control is feeding the test machine at up to 600 $V_{DC}$. The dynamometer is operating at closed-loop speed control through a thyristor converter, feeding back power to the grid when the test machine is operating in motor mode.

**Table 6.** Critical measurement equipment.

| Quantity | Equipment |
| --- | --- |
| Torque | ETH DRBK-500-n |
| Speed, Position | Kübler 8.5020.2514.1500 |
| Voltage | LeCroy HVD3206A |
| Current | LeCroy CP150 |
| Power Analysis | LeCroy MDA805 |
| Mass flow | Swissflow SF800 |

## 5. Results

In this section the results from the measurements are presented and compared to the simulated values.

### 5.1. Machine Parameters, Back EMF and Torque Evaluation

The initial set of tests performed on the machine is to extract the main equivalent circuit model parameters. Firstly, the DC resistance of each coil is measured using a 4-terminal method resulting in a phase resistance of 62 mΩ. Secondly, the machine is connected with phase A in series with the parallel of phase B and C. The inductance seen with this type of connection needs to be scaled by 2/3 to obtain the synchronous inductance. By moving the rotor in small mechanical angle steps and locking it, the inductance for each position can be measured using two methods:

- LCR tester at 1 kHz
- voltage step response from a fast power supply with a final value at 19 A. The 20–80% rise time is extracted with an oscilloscope and re-scaled to a first order time constant

The results of the two measurement methods, compared to the predicted FEA synchronous inductance as a function of the mechanical position, are presented in Figure 12. The maximum and minimum values of the curves in Figure 12 represent the q-axis and d-axis inductance respectively. The machine presents a saliency $L_q/L_d = 1.45$ given by the V-shaped rotor geometry. The difference found between the two inductance measurement methods is likely due to the difference in relative permeability of the iron lamination which is both frequency and polarization dependent. In particular the relative permeability of iron at very low excitation current (LCR method) is typically lower compared to the one found with excitation values corresponding the region where the machine typically operates (step method) which directly affects the inductance.

Figure 13 shows the no-load back-EMF at 1016 rpm, with both FEA and measurements. The measured back-EMF presents some harmonics which are not found in the FEA, the source of these is unknown, but likely these are due to the effect of laser cutting on the lamination edges, uneven magnetization of the PMs installed in the machine and/or small stator/rotor eccentricities.

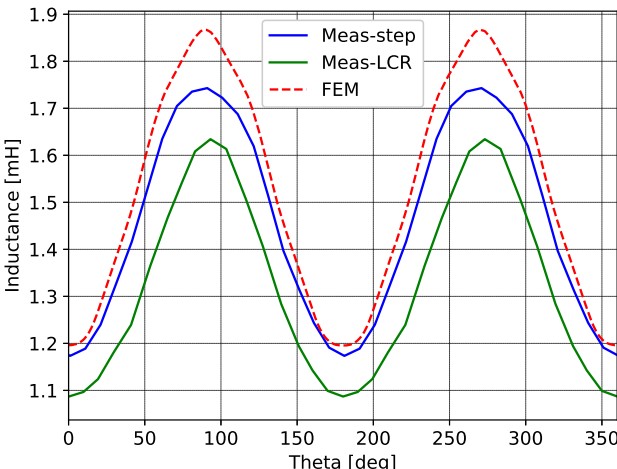

**Figure 12.** Inductance versus electrical position by FEA and two different measurement methods.

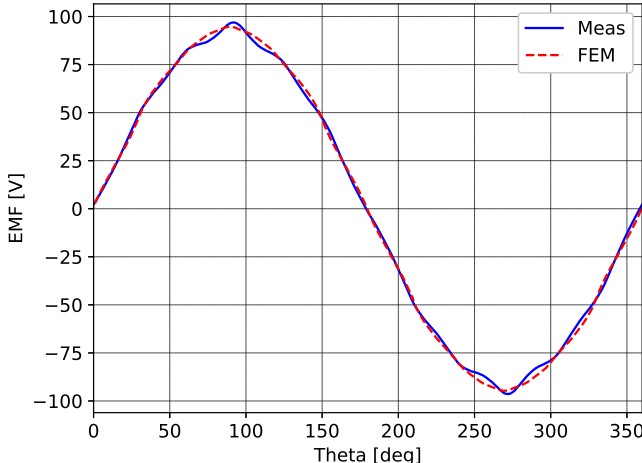

**Figure 13.** No-load back-EMF at 1016 rpm measured and simulated line-line voltage in Y-connection.

The torque is measured, with the torque sensor, in two different operating conditions, pure q-axis current (90 deg) and the MTPA angle at maximum current (115 deg). The results at 150 rpm are presented in Figure 14. A current value of 100 A RMS corresponds to a copper current density of 25 A/mm$^2$ which can be kept continuously for this machine as shown in [26].

### 5.2. Water Jacket Cooling Comparison

If the machine would have been equipped with a standard water jacket cooling, a maximum continuous current density of 10 A/mm$^2$ could instead be expected [7,24]. By removing the cooling channels from the slot area, and maintaining the same fill factor, the slots could theoretically fit 40% more copper. Altogether, the maximum allowed continuous phase current would then be 56 A instead of 100 A, corresponding to 62 Nm output torque instead of 110 Nm, according to Figure 14. Overall, by using direct oil cooling, the continuous power of the machine, for the same operating speed, increases by over 75%.

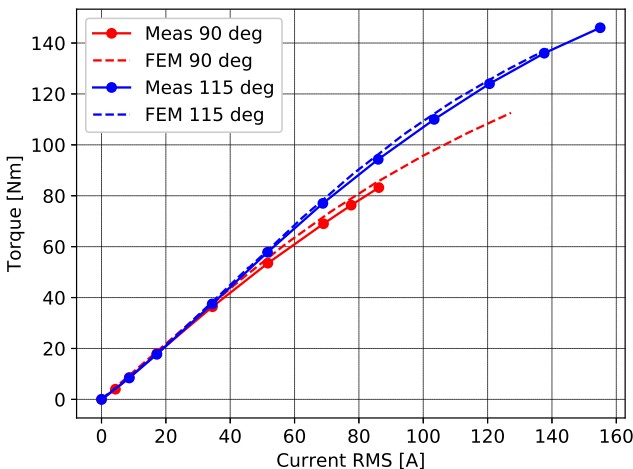

**Figure 14.** Average torque output at 150 rpm rotor speed, 90 and 115 deg current angle. Measurement and simulation comparison.

### 5.3. Calorimetric Evaluation of Efficiency

Electromagnetic efficiency through FEA, not including mechanical losses, is presented in Figure 15. The six operating points where the efficiency of the machine has been measured with the calorimetric setup are also marked in the figure with letters. The calorimetric method is presented in (8), and the measured efficiency is calculated using the total AC electrical active power $P_{in}$ from the power analyser as:

$$\eta = \frac{P_{in} - P_{cal}}{P_{in}} \ . \tag{9}$$

The output mechanical power is defined as $P_{mech} = \omega\,\tau$ . The measured efficiencies are presented in Table 7 and summarized in Figure 16. The measured efficiency is within 0.9% of the simulated one. The difference between the two can be attributed to two main reasons:

- Mechanical losses, such as friction in the bearings and windage losses, which are not included in the FEA efficiency but are present in the measurements
- Leakage heat towards ambient air and through the shaft. Although, all possible precautions have been taken to minimize the heat leakage it is not possible to completely remove this source of error

**Table 7.** Results from calorimetric measurements.

| Point | $\omega$ (rpm) | $\tau$ (Nm) | $i$ (A RMS) | $P_{mech}$ (kW) | $P_{in}$ (kW) | $P_{cal}$ (W) | $\eta$ (%) | $\eta_{FEA}$ (%) | $\eta_{FEA}$-$\eta$ (%) |
|---|---|---|---|---|---|---|---|---|---|
| A | 1003 | 27.5 | 30.0 | 2.89 | 3.16 | 233 | 92.63 | 92.8 | 0.17 |
| B | 1005 | 56.0 | 60.0 | 5.89 | 6.65 | 660 | 90.07 | 90.8 | 0.73 |
| C | 1985 | 26.6 | 30.0 | 5.52 | 5.97 | 370 | 93.80 | 94.3 | 0.50 |
| D | 1990 | 56.6 | 60.0 | 11.78 | 12.65 | 860 | 93.20 | 94.0 | 0.80 |
| E | 3004 | 27.0 | 30.0 | 8.49 | 9.05 | 570 | 93.70 | 94.6 | 0.90 |
| F | 3007 | 57.0 | 60.0 | 17.96 | 18.74 | 785 | 94.66 | 94.9 | 0.24 |

The mechanical power can also be used to estimate the efficiency, however the accuracy of the torque sensor reading is lower compared to the input power reading and would lead to a higher uncertainty. Measurements above base speed are unfortunately not possible with the current dynamo setup.

In Figure 17 the calorimetric run for point E is presented. It is of great importance to reach thermal steady state in order to achieve an accurate calorimetric reading.

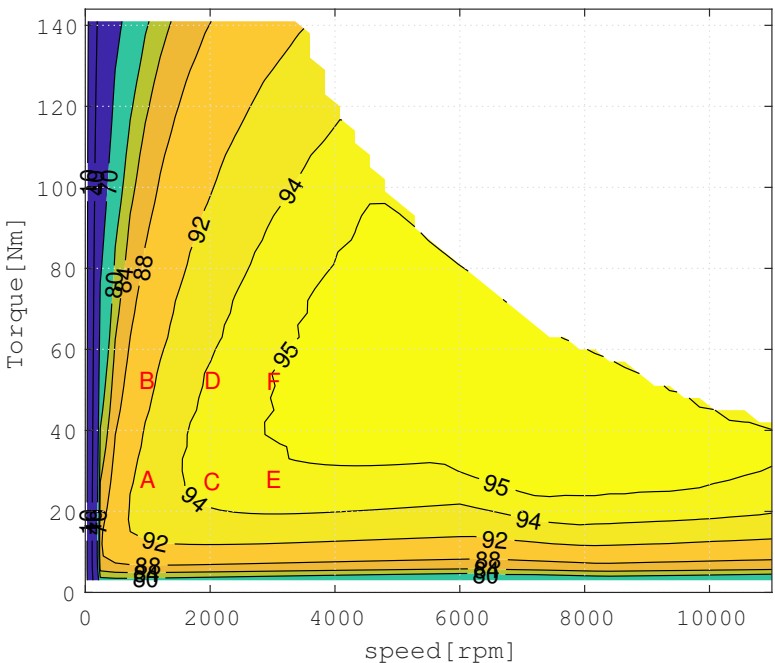

**Figure 15.** Efficiency map established with FEA while operating with a MTPA control strategy. Letters A-F represent the points verified with calorimetric measurements presented in Table 7.

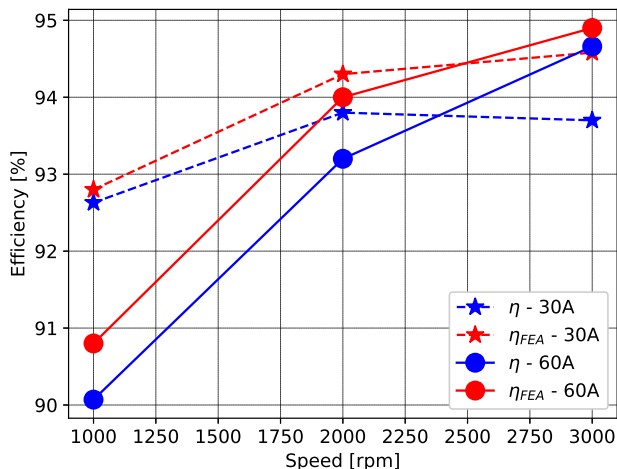

**Figure 16.** Measured and simulated efficiency comparison. $\eta$ is measured efficiency, $\eta_{FEA}$ is efficiency established with FEA.

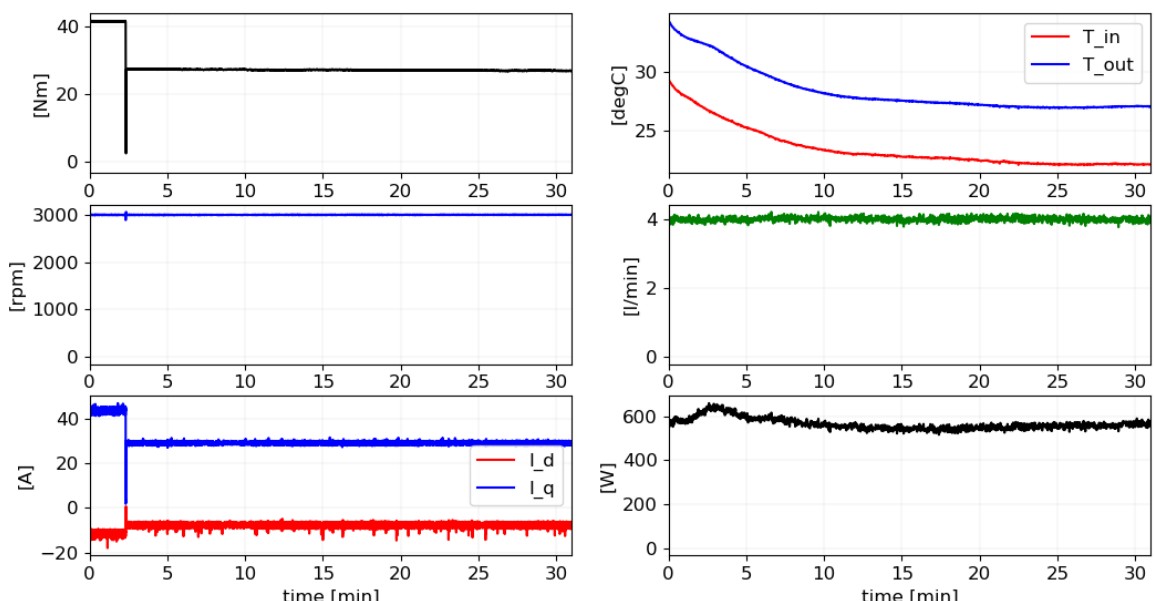

**Figure 17.** Calorimetric run at 3000 rpm and 30 A RMS. The plots are showing: top left output torque, top right oil inlet and outlet temperature, middle left mechanical speed, middle right oil flow rate, bottom left d-axis and q-axis currents, bottom right calorimetric loss.

## 6. Conclusions

This paper presents the design of a tooth coil winding PMSM machine for traction application, focusing on the electromagnetic design and performance verification. The solution adopted integrates stator cooling, both thorough the slot and the stator yoke. Original elements of this machine design are direct-oil cooling channels positioned in the stator yoke without negatively affecting the electromagnetic performance, and direct in-slot cooling by using a thermally conductive epoxy resin to create oil channels within the slot. It is shown that for the Q12p10 machine, the electromagnetic performance is negligibly affected when removing iron in the stator yoke for cooling channels if the position is carefully selected to the yoke area between the phase windings. The machine is designed such that it is possible to use a linear winding machine to pre-wind the coils on a bobbin, potentially leading to a reduced manufacturing cost for high volume production.

The adopted cooling solution enables a continuous copper current density of up to 25 A/mm$^2$ [26]. This allows for a 75% higher output torque and power density comparing with a corresponding water-jacket-cooled machine. Inductance, torque and no-load back EMF are measured and compared with FEA results, showing very good agreement. Furthermore, the efficiency of the machine is validated in one operating point, using a calorimetric direct-loss measurement set up, matching with FEA within 0.9 units of percent. The peak efficiency according to FEA is a wide area above 95%, which is on par state-of-the-art automotive traction machines for higher power levels. The overall net power density (19 kW/l) is comparable with the current state-of-the-art traction machines on the market, even with higher power ratings, despite this machine being an early prototype. Further design improvements towards series production are also likely to bring the gross power density to very appealing levels by minimizing the volumetric overhead of coolant interfaces and connection terminals.

**Author Contributions:** Conceptualization, A.A.; methodology, A.A. and S.S.; software, A.A. and S.S.; validation, A.A. and S.S.; formal analysis, A.A. and S.S.; investigation, A.A. and S.S.; resources, A.A. and S.S.; data curation, A.A. and S.S.; writing—original draft preparation, A.A. and S.S.; writing—review and editing, A.A., S.S. and E.G.; visualization, A.A. and S.S.; supervision, E.G and T.T.; project administration, T.T.; funding acquisition, T.T. All authors have read and agreed to the published version of the manuscript.

**Funding:** Research funded by Swedish Energy Agency and the Swedish Governmental Agency for Innovation Systems (VINNOVA)

**Acknowledgments:** Sincere gratitude to Bevi AB for sharing valuable experience on manufacturing challenges and assembling the machine prototypes.

**Conflicts of Interest:** The authors declare no conflict of interest. The funders had no role in the design of the study; in the collection, analyses, or interpretation of data; in the writing of the manuscript, or in the decision to publish the results

## Abbreviations

The following abbreviations are used in this manuscript:

PMSM    Permanent magnet synchronous machine
HTC     Heat transfer coefficient
FEA     Finite element analysis
TCWM    Tooth coil winding machine
DW      Distributed winding
MMF     Magneto-motive force
FW      Field weakening
SVM     Space vector modulation

## Appendix A

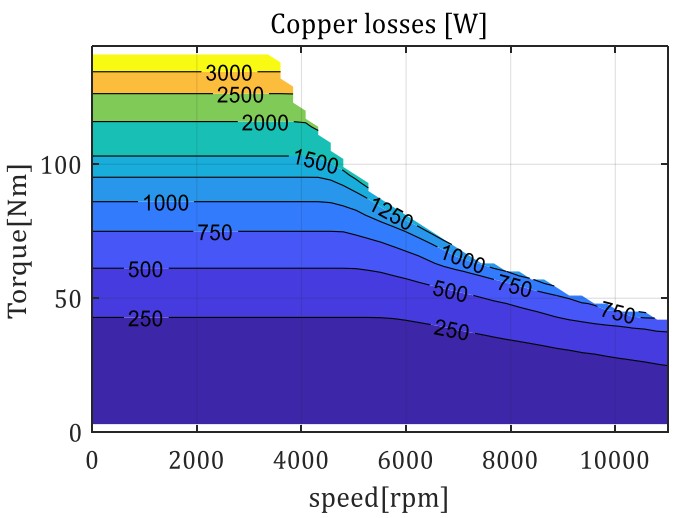

**Figure A1.** Copper loss map from FEA.

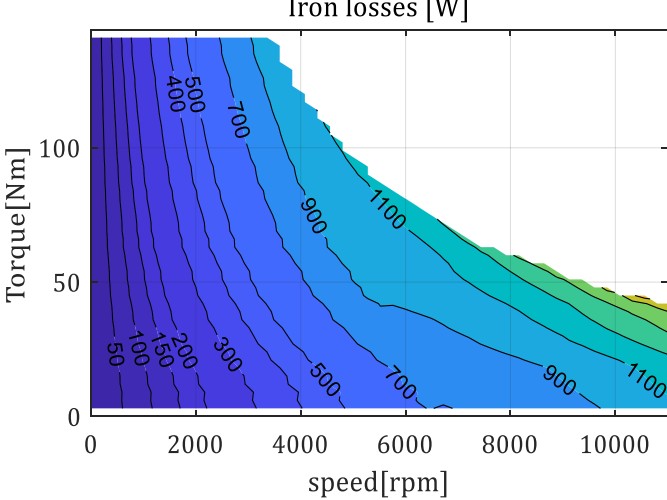

**Figure A2.** Iron loss map from FEA.

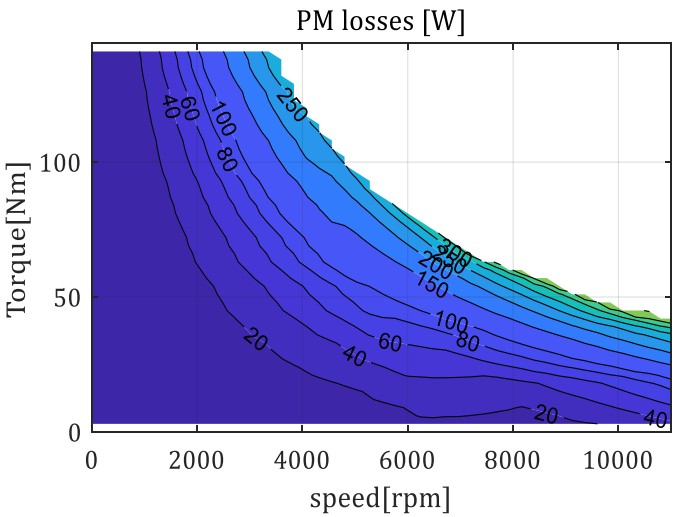

**Figure A3.** PM loss map from FEA.

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
