# Peer review of "Electromagnetic and Calorimetric Validation of a Direct Oil Cooled Tooth Coil Winding PM Machine for Traction Application"

_energies, doi:10.3390/en13133339_

Round 1
Reviewer 1 Report
This paper studies a 50kW, 11000rpm high voltage oil-cooling electric vehicle driving motor, which directly cools the motor by forming a cooling channel in the stator yoke and slot, which greatly improves the heat dissipation effect of the motor and provides experimental data, It has practical application significance.
- why you make a 600V motor, what is the basis. As far as I know, BMW and Toyota are gradually lowering the voltage level to ensure passenger safety.
- How does the author consider the chemical reaction of the cooling oil with the insulation or potting material, can it be effectively cooled for a long time.
- Please provide the mass of each part of the motor, and calculate the motor power mass density and power volume density. In addition, please provide the length of the winding end.
- What about the motor reluctance torque? From the results of Fig. 8 and Fig. 13, the motor has obvious reluctance torque, and the author needs to supplement the proportion of reluctance torque at each current density
- In Table 4, which part of "Barrier" refers to? Please mark in the figure. In addition, it is best to clearly explain which base value of the comparison is. The text part needs to be added.
The yoke has a cooling channel. The author claims on line 144 of Page 6 that it occupies 77% of the space of the yoke. What happens to the saturation change of the yoke at different current densities? And the flow output needs to be given, and the pressure data also needs to be analyzed.
- In Figure 5, the frozen magnetic permeability method should be added to consider the change of the magnetic field when the permanent magnetic field exists
More able to reflect whether the design at this time is reasonable
- Please analyze the end effect and flux leakage
- The cause of the diversion in Fig. 11 needs to be analyzed. The same situation appears in Fig. 12. The back-EMF harmonics of the prototype are more serious, please explain.
- In the analysis of loss, how to consider AC loss, especially at higher frequencies and current amplitude. In Figure 14, the author did not analyze the impact of driver harmonics on losses. Details of the losses caused by PWM harmonics need to be added.
- In the article, I don’t see any data that can support the author ’s statement that the motor can run continuously at 25A / mm2. The experimental data in FIG. 6 only achieved 15A / mm2, and the time lasted about 2.5min. Please provide evidence data or previous the work did. It is not appropriate to put it directly in the conclusion
Author Response
Dear Reviewer,
The authors would like to thank the reviewers for their work and the valuable comments. Hopefully, we have managed to treat them in such a way that this version of the paper is improved:
The improved sections in the manuscript are highlight with different font colors.
The point-by-point response is found in the attached pdf.

Reviewer 2 Report
The authors have developed and described a significant and complete work for the project of a PM synchronous traction motor. This paper deserves, in my opinion, to be included in the special session. However, some minor points need clarification to the reader to fully "appreciate" their work. Follow, I list these points.
1) Lines 99 - 101: Insert in a table a list of all values for the machine geometrical and load parameters.
2) Figure 2: Lacks a table where the values considered for volume, air gap flux density, and current density are listed to the reader.
3) Line 131: Please, specify the electromagnetic characteristics of the ferromagnetic material used, lamination thickness, and how they were cut.
4) Lines 136 - 138: Concerning the phrase "...The reluctance change for coil self...to be negligible." This needs to be verified even only using a FE analysis for with and without cooling channels.
5) Lines 143-144: Concerning the choice of cooling channels thicknesses show, in terms of percentage change concerning the PM flux, how changed for the parametric sweep values.
6) Line 144: As stated in "...As a reference, 2.0mm is used...". Please, clarify in the manuscript what means "reference" in that context. - It is also essential to explain how the width of the channels was selected. Why the two have different widths?
7) - Identify the oil and its thermal characteristics.
8) Line 238: Please, correct or clarify the following text "...ca 19 A.."
9) Lines 240-241: Concerning the results in Figure 11, one verifies non-neglectable differences between the two measurement techniques, and also from these and the FEM results. This needs a discussion and clarification to the reader, pointing which Ld and Lq values will be considered and why.
10) Line 244: Clarify why using a speed of 1016 rpm?
11) Lines 246-247: Concerning Figure 13, only it is shown results for 150 rpm. Questions that need clarification: why 150rpm? Why did you not test for an extensive speed range?
Author Response
Dear Reviewer,
The authors would like to thank the reviewers for their work and the valuable comments. Hopefully, we have managed to treat them in such a way that this version of the paper is improved:
The improved sections in the manuscript are highlight with different font colors.
A point-by-point response to comments can be found in the pdf attached.

Reviewer 3 Report
The introduction of the paper is well written and covers all the background.
The Machine design section can be improved by writing a few equations rather than simply citing the sources. In that way, a reader could get a complete overview of designing as well as the cooling method technique introduced in this paper. However, this should be taken as a suggestion (optional revision).
The paper is acceptable in its current form. The authors should check for minor spellcheck before the final publication.
Good Luck.
Author Response
Dear Reviewer,
Thank you for your feedback. We have added further explanations in the machine design sections and performed a spellcheck and reviewed the wording.